

# Choline regulation of triglycerides synthesis through ubiquintination pathway in MAC-T cells

Mengxue Hu and Lily Liu

College of Life Science, Southwest Forestry University, Kunming, Yunnan Province, China

## ABSTRACT

This study aims to investigate the regulatory mechanism of choline (CH) on triglyceride (TG) synthesis in cows, with a specific focus on its potential association with high milk fat percentage in the gut of the Zhongdian yak. By employing combined metagenomics and metabolomics analysis, we establish a correlation between CH and milk fat production in yaks. Bovine mammary epithelial cells (MAC-T) were exposed to varying CH concentrations, and after 24 h, we analyzed the expression levels of key proteins (membrane glycoprotein CD36 (CD36); adipose differentiation-related protein (ADFP); and ubiquintin (UB)), cellular TG content, lipid droplets, and cell vitality. Additionally, we evaluated the genes potentially related to the CH-mediated regulation of TG synthesis using real-time qPCR. CH at 200 µM significantly up-regulated CD36, ADFP, UB, and TG content. Pathway analysis reveals the involvement of the ubiquitination pathway in CH-mediated regulation of TG synthesis. These findings shed light on the role of CH in controlling TG synthesis in MAC-T cells and suggest its potential as a feed additive for cattle, offering possibilities to enhance milk fat production efficiency and economic outcomes in the dairy industry.

## INTRODUCTION

Choline (CH), an essential nutrient, is known to play a vital role in various physiological processes, including lipid metabolism (*Potts et al., 2023*). Its significance in dairy production has been highlighted through the established relationship between CH and high milk fat percentage in Zhongdian yaks (*Liu et al., 2022*). However, despite this knowledge, the regulatory pathway of CH in milk fat or triglyceride (TG) synthesis remains unexplored.

The ubiquitination pathway is a crucial cellular process responsible for protein degradation and turnover (*Mello-Vieira, Bopp & Dikic, 2023*). Notably, it also holds implications in regulating lipid metabolism by targeting key enzymes and transcription factors involved in lipid synthesis and storage (*Schulman & Harper, 2021*). Since triglycerides are the primary component of milk fat, synthesized mainly by mammary epithelial cells, understanding how the ubiquitination pathway affects milk fat production through the regulation of milk fat-related proteins and rate-limiting enzymes is essential.

Corresponding author
Lily Liu, liulily0518@163.com

In light of this, our research plan involves culturing MAC-T cells in different concentrations of CH solution to investigate its impact on bovine milk fat production and its potential association with the ubiquitination signaling pathway. This study aims to review the current literature on the relationship between CH, ubiquitination, and lipid metabolism, focusing on recent studies that explore the effects of CH on triglyceride synthesis in MAC-T cells. Furthermore, we will present our own findings regarding the potential role of CH in promoting lipid synthesis through the ubiquitination pathway, discussing the implications of these discoveries for a better understanding of CH's involvement in lipid metabolism and its potential therapeutic applications.

## MATERIALS AND METHODS

### Reagent

The immortalized bovine mammary gland epithelial cell lines, known as MAC-T cells and spanning 40 generations, were generously provided by the laboratory at the Kunming Institute of Zoology, Chinese Academy of Sciences. Essential laboratory reagents included high glucose Dulbecco's modified Eagle's medium (DMEM, Gibco™ 11965092, 99%), fetal bovine serum (FBS, Gibco™ 10099–141, 98%), and penicillin-streptomycin (Gibco™ 15140122, concentrated: 100×), all sourced from Thermo Fisher Scientific (Waltham, MA, USA). Specific antibodies used in this study were CD36 (No. ab252922) and ADFP (No. a181452) obtained from Abcam (Cambridge, UK), ubiquitin (UB) antibody (Ubiquitin, sc53509) purchased from Santa Cruz (Dallas, Texas, USA), Beta-actin antibody (66009-Ig) acquired from Protein tech (San Diego, California, USA), and HRS-conjugated secondary antibody (7076S) from Cell Signaling Technology (Danvers, MA, USA). Additionally, CH powder (CAS: A39755IP, 99%) was procured from Thermo Fisher Scientific (Waltham, MA, USA), and the Cell Counting Kit-8 (96992) was obtained from SIGMA (St. Louis, MO, USA).

### Cell culture and addition of CH solution

MAC-T cells were cultured in DMEM complete medium, which was supplemented with 10% FBS, 100 U/mL penicillin, and 100 µg/mL streptomycin. The cells were maintained in a constant temperature incubator at 37 °C with 5% $CO_2$, and the culture medium was refreshed every 2–3 days. Subculturing was performed when the cell density reached 70% and 80%, indicating favorable growth status. For subculturing, the cells were seeded at a density of $2.5 \times 10^5$ cells per well in a six-well plate. The old culture medium was aspirated, and the cells were washed once with PBS. Subsequently, 0.25% trypsin was added for digestion, and after centrifugation and removal of the supernatant, fresh complete culture medium was added and gently mixed. The cells were then transferred to the 6-well plate for continued culture to be used in subsequent experiments.

MAC-T cells were maintained in DMEM complete medium, supplemented with 10% FBS, 100 U/mL penicillin, and 100 µg/mL streptomycin. The cells were cultured in a constant temperature incubator at 37 °C with 5% $CO_2$, and the culture medium was refreshed every 2–3 days. Subculturing was initiated as previously described in *Hu et al. (2023)*. To investigate the effects of CH, CH powder was incorporated into an appropriate

amount of cell culture medium and filtered through a sterile filter to remove any insoluble particles. The resulting culture medium contained various concentrations of CH (100, 150, 200 µmol/L), and this medium was used to incubate MAC-T cells for 24 h. Following the incubation period, total cellular proteins were extracted, quantified, and prepared for further analysis.

## Total protein extraction and western blotting

The protein extraction, quantification, and western blot procedures were conducted following the methodology outlined in a previous study (*Hu et al., 2023*). Briefly, proteins were extracted using RIPA buffer, quantified *via* BCA method, and separated on a 10% SDS-PAGE gel. After electrotransfer to PVDF membrane, primary antibodies were applied and followed by appropriate secondary antibody incubation. Grayscale analysis of protein bands was performed using Image J after ECL chemiluminescence detection.

## Nile red staining and triglyceride content assay

Nile red staining and TG content assay for MAC-T cells were performed following our previous procedure (*Hu et al., 2023*). Briefly, cells were seeded in a 12-well plate and incubated with CH in the culture medium for 24 h. After fixation and staining with Nile red and DAPI, lipid images were captured using a confocal microscope. TG content was assessed using a colorimetric assay as procedure of TG detection kit from Nanjing Jiancheng Bioengineering Institute (Nanjing, China).

## Cell viability assay

Cell viability of MAC-T cells was assessed using the CCK-8 method. MAC-T cells were seeded in a 96-well cell culture plate and cultured until they reached 70% confluence. Next, CH solution was added to the cells, and they were incubated for 24 h. Following the incubation period, the culture medium was replaced with 10% CCK-8 basic medium, and the cells were further incubated for 3 h.

To determine cell viability, the absorbance at 450 nm for each cell group was recorded using an automated microplate reader. The calculation of cell viability was based on the absorbance values derived from the CCK-8 assay.

## Detection of mRNA expression levels of relevant genes

A comprehensive selection of 37 genes associated with lipid synthesis, ubiquitin-proteasome, and ubiquitin-lysosome signaling pathways was meticulously curated using data from NCBI (https://www.ncbi.nlm.nih.gov/) and KEGG websites (http://www.genome.jp/kegg/kegg2.html). To assess mRNA expression levels for these chosen genes, quantitative real-time PCR (qPCR) was performed with GAPDH serving as the reference gene.

For the qRT-PCR analysis, Vazyme SYBR Green Mix was employed for quantitative expression analysis, and the reaction system and program followed the guidelines specified in the SYBR Green Mix manual. Amplification was performed using the ABI QuantStudio five fluorescence quantitative PCR instrument.

To calculate the relative expression levels of the target gene mRNAs, the $2^{-\Delta\Delta Ct}$ method was applied, where $\Delta\Delta Ct$ represents the difference in Ct values between the treatment group and the control group ($\Delta Ct$ treatment group–$\Delta Ct$ control group). This method enables the accurate assessment of changes in gene expression under the experimental conditions.

## Data analysis

All experimental data were presented as mean ± standard deviation, and each experiment was independently replicated three times to ensure the reliability of the results. To assess the significance of differences between two groups, independent sample t-tests were performed using SPSS software (Version 21.0).

For statistical significance, $P < 0.05$ was considered statistically significant, indicating that the observed differences were unlikely due to random chance. Moreover, $P < 0.01$ was considered highly significant, signifying a more robust and pronounced difference between the compared groups. These criteria were used to determine the statistical significance of the experimental results.

# RESULTS

## Effects of different concentrations of CH on the expression of lipid synthesis-related proteins and ubiquitin proteins in MAC-T cells

MAC-T cells were cultured with various concentrations of CH, and the expression levels of lipid synthesis-related proteins and UB were analyzed (Fig. 1). The results revealed significant changes in the expression levels of key proteins involved in lipid metabolism. Specifically, the fatty acid transporter protein CD36 exhibited a significant upregulation of 2.78-fold ($P < 0.05$) when MAC-T cells were cultured with 200 μM CH. However, lower concentrations of CH, such as 100 and 150 μM, led to a notable decrease in CD36 expression by 17.03% ($P < 0.05$) and 27.26% ($P < 0.05$), respectively. Similarly, the lipid droplet marker protein ADFP showed an increase in expression by 1.20, 1.15, and 1.25-fold ($P < 0.01$) in MAC-T cells cultured with 100, 150, and 200 μM CH, respectively. Furthermore, the levels of ubiquitinated proteins (UB) were significantly elevated (2.06-fold, $P < 0.01$) in MAC-T cells exposed to 200 μM CH, indicating a potential involvement of the ubiquitination signaling pathway in mediating CH's effect on lipid synthesis. These findings collectively suggest that a CH concentration of 200 μM notably enhances the expression of lipid synthesis-related proteins in MAC-T cells. This implies that CH might modulate lipid synthesis in MAC-T cells, possibly through the ubiquitination signaling pathway.

## Effects of different concentrations of CH on cell viability and trigly- ceride synthesis in MAC-T cells

To investigate the impact of different concentrations of CH on MAC-T cells, we conducted a series of assays to evaluate cell viability and triglyceride synthesis. The CCK-8 assay (Fig. 2A) revealed that all CH concentrations (100, 150, and 200 μM) significantly promoted MAC-T cell viability compared to the control group. Among these
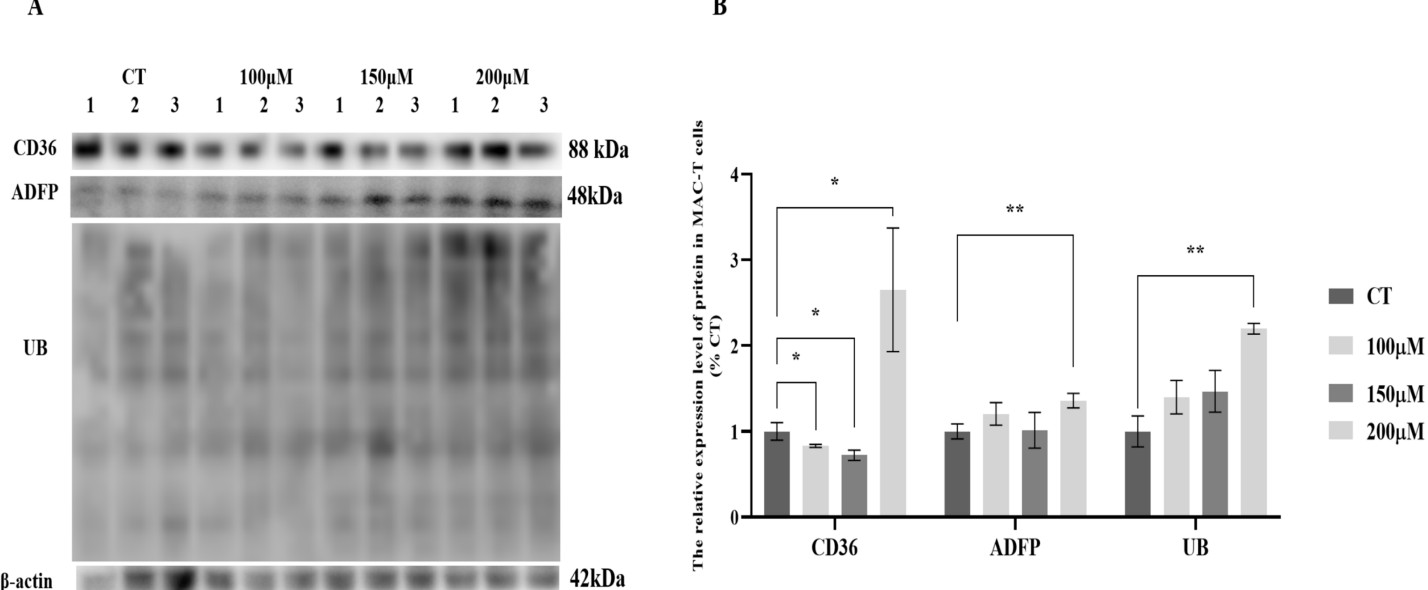

**Figure 1 Effects of choline at different concentrations on the expression levels of CD36, ADFP and UB proteins in MAC-T cells.** * Denotes $P < 0.05$, ** Denotes $P < 0.01$. (A) Western blot analysis of the protein levels of CD36, ADFP, and UB in MAC-T cells; (B) The relative protein levels of CD36, ADFP, and UB in MAC-T cells.

concentrations, 200 μM CH exhibited the most pronounced effect in enhancing cell viability. Additionally, CH at the concentration of 200 μM also had a notable impact on cell viability. Furthermore, we quantified the TG content in MAC-T cells (Fig. 2B) and found that 200 μM CH induced a substantial increase in the total TG content within the cells, showing a 2.36-fold rise compared to the control group ($P < 0.05$). Nile red staining (Fig. 2C) further confirmed the impact of 200 μM CH on triglyceride synthesis, as MAC-T cells cultured with this concentration exhibited a higher number of larger lipid droplets. In summary, our findings demonstrate that 200 μM CH significantly enhances triglyceride content and lipid droplet size in MAC-T cells. These observations highlight the potential role of CH in modulating lipid metabolism in MAC-T cells, suggesting its significance in dairy production and related biological processes.

## Regulation of relevant genes in MAC-T cells by CH

To elucidate how CH influences TG synthesis in MAC-T cells through the ubiquitination signaling pathway, fluorescence quantitative PCR was utilized to analyze the relative expression levels of 37 relevant genes in MAC-T cells treated with 200 μM CH, using GAPDH as the reference gene. The results are presented in Table 1, revealing intriguing insights into the impact of CH on gene expression. In the context of the lipid synthesis pathway, we observed a significant upregulation ($P < 0.05$) in the expression levels of *LDLR*, CD36, *FASN*, *LPIN1*, and *ADFP* (*PLIN2*), which is consistent with the western blot findings. On the other hand, within the ubiquitination signaling pathway, the expression levels of *MAPK1*, *ARF6*, *STAM1*, *VPS45*, *CHMP2B*, *CHMP3*, *CYHR1*, *UBA52*, *UBA7*, *MX1*, and *RPS27A* were significantly downregulated ($P < 0.05$). However, this observation

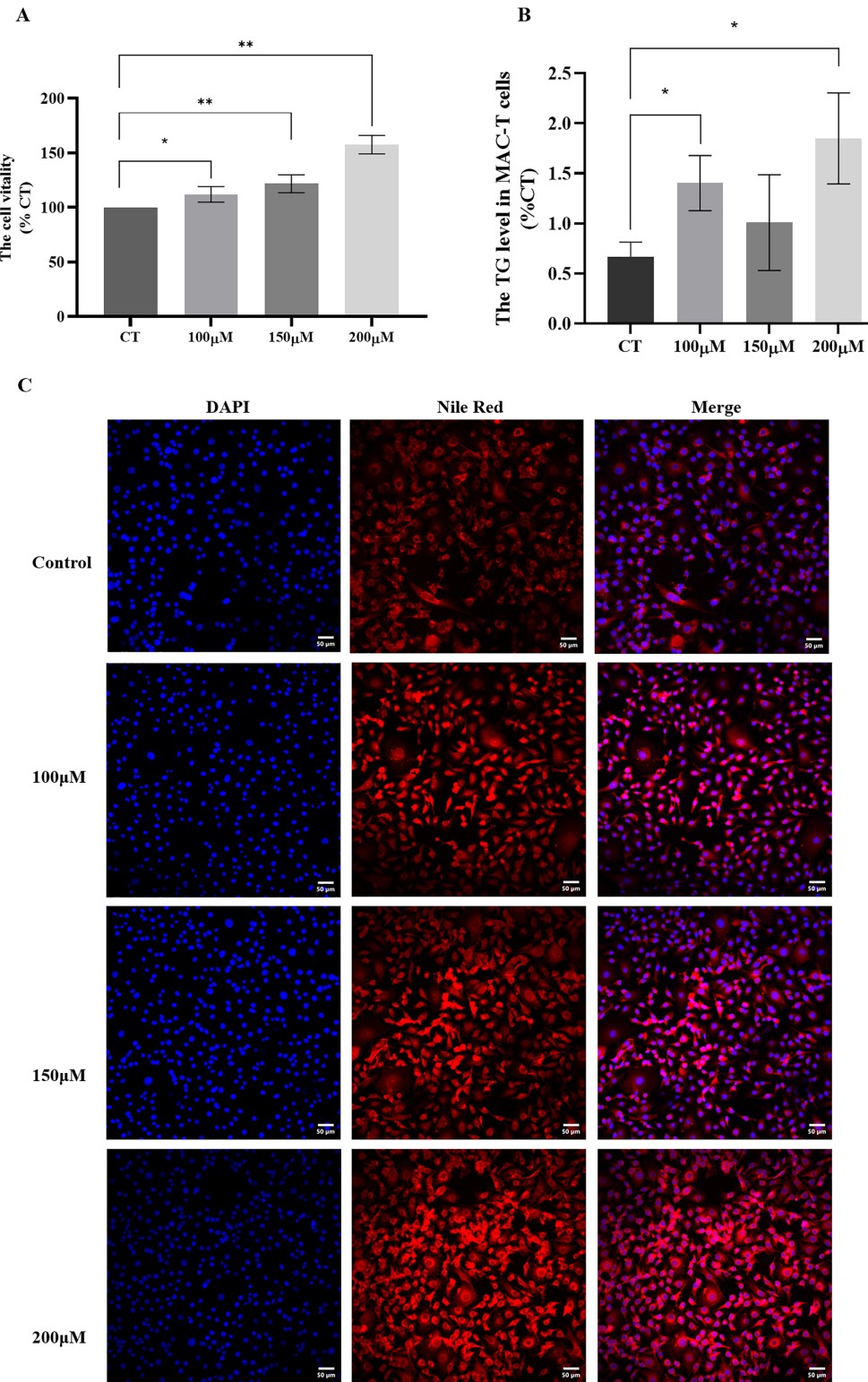

**Figure 2 Effects of different concentrations of choline on cell viability and triglyceride content in MAC-T cells.** * Denotes $P < 0.05$, ** Denotes $P < 0.01$. (A and B) The effects of different concentrations of choline on cell viability and triglyceride content in MAC-T cells; (C) Nile red staining of lipid droplets in MAC-T cells cultured with different concentrations of choline (50 µM).           
**Table 1  The information of primers of the candidate genes and their relative expression.**

| Gene | Name | Primer (5′→3′) | Relative expression |
|---|---|---|---|
| GAPDH | Glyceraldehyde-3-phosphate dehydrogenase | AGATGGTGAAGGTCGGAGTG<br>CGTTCTCTGCCTTGACTGTG | / |
| LPL | Lipoprotein lipase | AGCTCCAAGTCGCCTTTCTC<br>TCCTGGTTGGAAAGTGCCTC | 0.47 |
| LDLR | Low density lipoprotein receptor | TGTTGGACACACGTACCCAG<br>AAGGTCGCGACTTGTCTCAG | 1.14** |
| CD36 | Platelet glycoprotein 4 | GACGGATGTACAGCGGTGAT<br>GAAAAAGTGCAAGGCCACCA | 3.32* |
| ACSL1 | Acyl-CoA synthetase long-chain family member 1 | AGCCGCATTTCACTTTTACTGC<br>AGCTCTTTAGGGCAAACCCC | 0.49** |
| FABP3 | Fatty acid-binding protein 3 | ACGCGTTCTCTGTCGTCTTT<br>AACCGACACCGAGTGACTTC | 1.13 |
| ACACA | Acetyl-coa carboxylase/biotin carboxylase 1 | ACGGCTGACTGGAGTTGAAG<br>AACGTCTGCTTGTCCGTCTT | 0.61 |
| ACBP | Aacyl-coa-binding protein | TGGAATCTTTGCAACACCGC<br>TGTCACCCACAGTTGCTTGT | 0.77** |
| FASN | Fatty acid synthase | CCTCAAGATGAAGGTGGTGCT<br>GGCCCTGGGTTATATCGAGC | 1.64** |
| SCD | Stearoyl-CoA desaturase | TCCTGATCATTGGCAACACCA<br>CCAACCCACGTGAGAGAAGAA | 0.83* |
| DGAT1 | Diacylglycerol o-acyltransferase 1 | TACCCCGACAACCTGACCTA<br>GGGAAGTTGAGCTCGTAGCA | 0.85* |
| LPIN1 | Lipin 1 | CTTCGATTCCCAAACCGGGA<br>TCACAGTGACGAACACCTGG | 4.01* |
| ADFP<br>(PLIN2) | Perilipin-2 | GCGTCTGCTGGCTGATTTC<br>AGCCGAGGAGACCAGATCATA | 1.16** |
| APOE | Apolipoprotein E | CGGTTTCTGGAGGCGAAGAA<br>CTCCATATCCGCCTGGCATC | 0.68 |
| PRKCA | Calcium-activated, phospholipid- and diacylglycerol- dependent serine | GACTTCGGGATGTGCAAGGA<br>CGTACGGCTGATAGGCGATT | 0.96 |
| MAPK1 | Mitogen-activated protein kinase 1 | AACAAAGTCCGAGTCGCCAT<br>CGATGGTCGGTGCTCGAATA | 0.72** |
| ARF6 | ADP-ribosylation factor 6 | AACTGGTATGTGTCAGCCCTC<br>GAAAGAGGTGATGGTGGCGA | 0.56* |
| STAM1 | Signal transducing adapter molecule 1 | CCTGGTACTGCGGCTAACAA<br>ACGAACTTTCCGGCCTTCAT | 0.59** |
| EEA1 | Early endosome antigen 1 isoform x4 | CAGGCCCAGGACAGCTTAAA<br>GCAAGTTCCTGTGCTGCTTG | 0.93 |

(Continued)

**Table 1** (*continued*)

| Gene | Name | Primer (5′→3′) | Relative expression |
|---|---|---|---|
| HERC3 | HECT and RLD domain containing E3 ubiquitin protein ligase 3 | CTCGAGGGCCTAGCTGTCT TTTGTCAGAAGGGTCTGGCG | 0.92 |
| VPS45 | Vacuolar protein sorting 45 homolog | CCCCAAAGATGCTGTGGCTA AGTGTGCTGGGGCCTAGATA | 0.85** |
| CHMP2B | Charged multivesicular body protein 2b | ACGAGGTACACAGAGGGCTA AGCTGTTTGGCTAAAACTCTGC | 0.60* |
| CHMP3 | Charged multivesicular body protein 3 | GTTTGAAATCACCGCAGGGG CTAAAGGTTCAGGCTCCGGG | 0.64* |
| PIP5K | Phosphatidylinositol-4-phosphate 5-kinase, putative | CTCAGCACCTGGAAGAGCAA TTCTTCTTTCCCCGAGCCAC | 0.72 |
| CYHR1 | Cysteine and histidine rich 1 | GCCAACCTGCTTTTGGGAAG GGTTGTGAAAACGGCCACAA | 0.72* |
| CP | Ubiquitin-like domain-containing CTD phosphatase 1 | CATGGTGGCCAAAGGTGTTG CATCTGCTGGAGATTTTTGGCA | 1.15 |
| PSMC1 | proteasome 26S subunit, ATPase 1 | GGTACGACTCCAACTCAGGC ATCCGGTTTGTGGCCATGAT | 0.89 |
| PSMC3 | 26S protease regulatory subunit 6A | TGAACAAGACGCTGCCGTAT TGCCGCGTAGAGGTTTTGAT | 0.87 |
| PSMC5 | 26S proteasome regulatory subunit 8 | CTCTGCACAAGATCCTGCCT ATGCTTCACAGGCAGCTCAA | 0.96 |
| PSMD12 | 26S proteasome non-ATPase regulatory subunit 12 | ATACGTCAGGCATCTCGCAG GGCCATGTTGTAGGGGACAA | 1.04 |
| UBC | Ubiquitin-C | GGGAGGTGTTTTAAGTTCTCCCT TTGAACTCTAACCCACCCCTAAC | 0.61 |
| UBA52 | Ubiquitin-60S ribosomal protein L40 | GCCCAGTGACACCATTGAGA GCAGGGTGGACTCTTTCTGG | 0.73** |
| UBA7 | Ubiquitin-like modifier-activating enzyme 7 | TCAGCAGGATGGTCTGAGGA AGTTCCAATACCAGCACCCG | 0.84* |
| TUBA | Tubulin beta chain | GTCTACTCCTGTTGCCTGC AGGCATTGCCGATCTGGAC | 0.77 |
| ISG15 | Ubiquitin-like protein ISG15 | CCATCCTGGTGAGGAACGAC GTCTGCTTGTACACGCTCCT | 0.58* |
| MX1 | Interferon-induced GTP-binding protein Mx1 | TGCCAACTAGTCAGCACTACATT TGTACAGGTTGCTCTTGGACTC | 0.82** |
| SPP1 | Secreted phosphoprotein 1 | TCCGCCCTTCCAGTTAAACC GCTTCTGAGATGGGTCAGGC | 0.82 |
| RPS27A | Ribosomal protein S27a | TTTCGTGAAGACCCTGACGG GTCTTTGCTGGTCAGGAGGAA | 0.83** |

**Notes:**
* Denotes *P* < 0.05.
** Denotes *P* < 0.01.

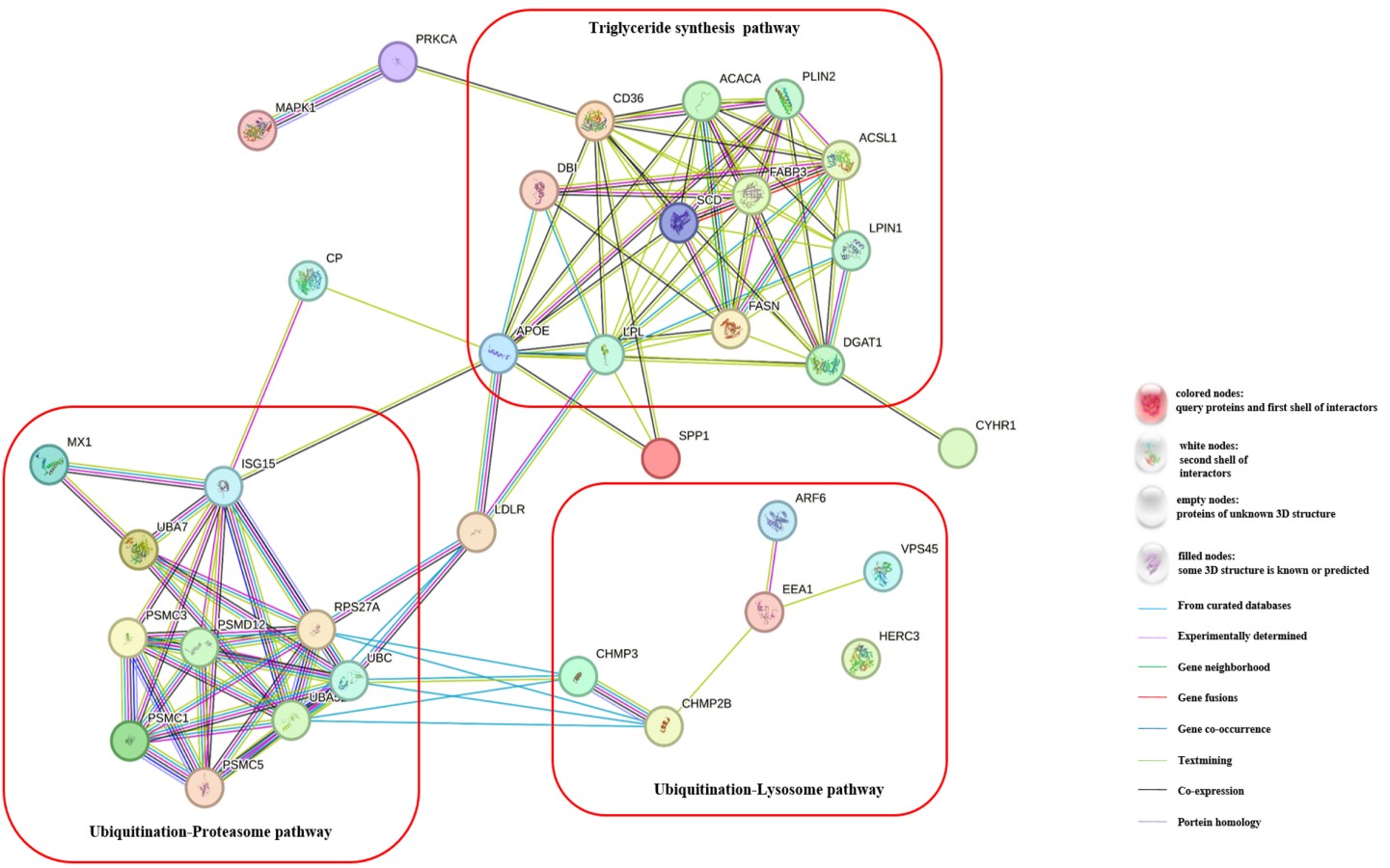

**Figure 3 Regulatory pathways of candidate genes affected by myristic acid in MAC-T cells.**

does not align with the western blot results, possibly due to feedback regulation mechanisms that influence gene expression.

To gain a deeper understanding of the effects of CH on TG synthesis, we constructed regulatory pathways of candidate genes using the STRING online software (https://string-db.org/), as depicted in Fig. 3. This analysis revealed that CH could impact the expression of genes involved in the ubiquitination-lysosome and ubiquitination-proteasome signaling pathways in MAC-T cells, ultimately modulating triglyceride synthesis. These results shed light on the complex regulatory mechanisms through which CH influences TG synthesis in MAC-T cells, involving multiple genes and signaling pathways. These findings offer valuable insights into the role of CH in lipid metabolism regulation, providing a basis for further research in the field of dairy production and related biological processes.

## DISCUSSION

TG constitute approximately 99% of the content of milk fat, making them the predominant component in this biological substance (*Liu & Zhang, 2020*). Consequently, changes in TG content within mammary epithelial cells directly reflect variations in milk

fat synthesis. Previous research has established a significant positive correlation between CH content in the intestines of Zhongdian yaks and milk fat percentage (*Liu et al., 2022*). Building on this finding, our current study delves into the function of CH at the mammary epithelial cell level, aiming to unravel its critical role and mechanism in regulating TG synthesis. By doing so, we aim to provide essential theoretical evidence for understanding the specific mechanisms and potential applications of CH in regulating milk fat synthesis.

MAC-T cells, a widely used bovine mammary epithelial cell line, serve as a crucial cell model for investigating mammary gland development and milk secretion. In mammary epithelial cells, TG synthesis occurs through two primary pathways (*Bionaz & Loor, 2008*). The first pathway involves the uptake of fatty acids from the external environment, facilitated by transport proteins on the cell membrane. Subsequently, these fatty acids undergo processes such as fatty acid activation and eventually get synthesized into TGs. The second pathway entails the direct utilization of small molecules, such as acetate, within the cells. This process is aided by enzymes like fatty acid synthase, leading to the synthesis of TGs. Understanding these pathways is essential for comprehending the intricate mechanisms of TG synthesis in mammary epithelial cells, providing valuable insights into milk secretion and mammary gland development.

CH is an essential nutrient with various biological functions, playing a crucial role in the health and normal physiological functions of all animals, including humans (*Van Parys et al., 2022*; *Beyer et al., 2021*) and ruminants (*Potts et al., 2023*; *Caprarulo et al., 2020*). Research has found that CH is involved in cell membrane structure (*Feng et al., 2020*), neurotransmission (*Giuliano et al., 2008*), phospholipid metabolism (*Ma et al., 2023*), lipid metabolism (*Arshad et al., 2023*), liver function (*Siciliani et al., 2023*), cell signaling (*Sivanesan et al., 2018*), and bile synthesis (*Kohjima et al., 2015*), among others (*Trousil et al., 2014*; *Sonkar et al., 2019*).

One of the primary sources of CH is through the breakdown of acetylCH by acetylcholinesterase, producing CH and acetic acid. Acetic acid is an important precursor in fatty acid synthesis. CH is an essential component of cell membrane phospholipids (*Feng et al., 2020*), participating not only in membrane synthesis and stability but also forming soluble CH precursors on the cell membrane (*Plagemann, 1971*). TG are composed of glycerol and three fatty acids, with glycerol being supplied by CH precursors. In the fatty acid synthesis and elongation pathways, fatty acids are initially derived from acetyl-CoA and then elongated stepwise through a series of enzymatic reactions. As a precursor of fatty acids, acetic acid is first activated to acetyl-CoA and then combined with a two-carbon compound (often acetyl-CoA) to form a shorter fatty acid chain. This chain is further elongated through a series of enzyme-catalyzed reactions. Studies have found that CH can be metabolized into methyl pyruvate, which can serve as a methyl donor for fatty acid synthesis (*Zeisel, 2017*). Therefore, CH can directly participate in TG synthesis by providing fatty acid precursors and engaging in fatty acid chain elongation by methylation and supplying glycerol.

Our research has also revealed that a 200 μM CH solution significantly increased the expression levels of enzymes related to *de novo* fatty acid synthesis and elongation, such as FASN, and ASCL1, in MAC-T cells. Additionally, the expression level of fatty acid transporter protein CD36 was significantly upregulated. We observed a significant increase in the triglyceride content and lipid droplet number in MAC-T cells cultured with 200 μM CH, further validating its direct involvement in TG synthesis. CD36 is a transport protein on the membrane of MAC-T cells, which can absorb long-chain fatty acids (LCFAs) and directly participate in the regulation of lipid intake (*Shen et al., 2021*; *Zhao et al., 2019*). As a transmembrane glycoprotein, CD36 contains multiple post-translational modification sites, such as glycosylation (*Hoosdally et al., 2009*), phosphorylation (*Kim et al., 2022*), palmitoylation (*Zhang et al., 2022*), acetylation (*Han et al., 2019*), and ubiquitination (*Zeng et al., 2023*), which regulate the stability, protein folding, and localization of CD36. Studies have found that CD36 can directly bind to ubiquitin proteins (*Zeng et al., 2023*; *Srikanthan et al., 2014*). In summary, the results of this study indicate that appropriate concentrations of CH could regulate fatty acid transport in MAC-T cells through the ubiquitination pathway, thereby promoting triglyceride synthesis.

To further determine whether CH can promote the synthesis of milk fat in MAC-T cells, this study further detected the expression level of ADFP, a marker protein for lipid droplets, as well as the content of triglycerides and lipid droplets in the cells. The results showed that 200 μM CH significantly increased the expression level of ADFP in MAC-T cells. ADFP is a surface protein on lipid droplets in cells, playing an important role in promoting the formation of intracellular lipid droplets (*Liu et al., 2012*). Studies have found that ADFP can directly bind to ubiquitin proteins and is regulated by the ubiquitination signaling pathway (*Masuda et al., 2006*; *Borbora, Rajmani & Balaji, 2022*). The increase in ADFP expression indicates an increase in the number of lipid droplets in the cells. This study indeed found that 200 μM CH significantly increased the TG content in MAC-T cells. After Nile red staining, it was also found that the content of lipid droplets in the cells increased significantly. This result also confirms that CH could indeed promote triglyceride synthesis in MAC-T cells, ultimately promoting milk fat production.

In summary, our investigation revealed that elevated concentrations of CH substantially heightened protein ubiquitination levels and the expression of genes related to ubiquitin in MAC-T cells. This suggests that there may be some association between CH and the ubiquitination signaling pathway. While there is no direct evidence of CH's involvement in ubiquitination.

Ubiquitination, a vital protein modification process in eukaryotic cells, involves the binding of small ubiquitin proteins to target proteins. This intricate process engages multiple enzymes, including E1 (activating enzyme), E2 (conjugating enzyme), and E3 (ligase), collaborating to attach ubiquitin to specific proteins (*Mello-Vieira, Bopp & Dikic, 2023*). Typically, ubiquitination flags a protein for degradation, guiding it to enter either proteasomes or lysosomes, thus maintaining cellular protein homeostasis (*Celada et al., 2023*; *Gu et al., 2023*). Additionally, ubiquitination plays a crucial role in regulating protein

subcellular localization, interactions, and functions, influencing diverse biological processes such as the cell cycle, DNA repair, and apoptosis (*Voss et al., 2021*; *Li, Li & Wu, 2022*).

To explore whether CH could regulate fatty acid transport through the ubiquitination pathway, this study further detected the expression level of UB in the cells. The results showed that 200 μM CH solution could significantly increase the expression level of UB in the cells. Additionally, we also found the three concentrations of CH all significantly increased cell viability. This result confirms the association between CH and the ubiquitination signaling pathway.

Besides being a nutrient, CH is also an important neurotransmitter, especially in transmitting signals between nerve cells (*Giuliano et al., 2008*). Its role is mainly achieved by binding to acetylcholine receptors on the nerve cell membrane. As acetylcholine receptors are membrane proteins, their structure and function are directly regulated by phosphorylation. Furthermore, some signaling proteins and enzymes involved in CH signal transduction can also be phosphorylated, regulating signal transmission and cell responses. In the processes of CH synthesis, release, degradation, and binding to acetylcholine receptors, some protein kinases and protein phosphatases, such as ACACA and FASN, can undergo phosphorylation and ubiquitination-like (*Ito et al., 2021*; *Kuang et al., 2020*), like ISG15, resulting in cross-regulation between phosphorylation and ubiquitination-like, contributing to cell signaling and metabolism (*Perng & Lenschow, 2018*). Particularly, in our previous study, we found that both ubiquitination and ISGylation are directly involved in TG synthesis (*Liu & Zhang, 2020*). In this study, we observed significant changes in the expression levels of phosphorylation-related genes, such as PRKCA and MAPK, as well as significant regulation of ISG15, a gene involved in ISGylation, and genes related to the ubiquitination signaling pathway, such as *UBA52*, *UBA7*, *PSMC3*, *PSMC5*, *VPS45*, *STAM1*, *CP*. Thus, CH could affect the expression and activity of genes related to fatty acid and TG synthesis by modulating the ubiquitination signaling pathway in MAC-T cells.

## CONCLUSION

By exposing MAC-T cells to various concentrations of CH solution, we observed significant effects on cellular processes related to TG synthesis. Specifically, a 200 μM CH treatment led to a substantial increase in the expression levels of key proteins involved in TG synthesis, including the fatty acid transport protein CD36, the surface protein of lipid droplets ADFP, and the ubiquitin protein UB. In summary, our study elucidates the crucial role of CH in regulating TG synthesis in mammary epithelial cells. CH's involvement in providing fatty acid precursors and participating in fatty acid elongation directly impacts milk fat production. Additionally, the potential associations with the ubiquitination and phosphorylation signaling pathways add further depth to our understanding of CH's multifaceted functions in cellular processes. Our findings provide important theoretical evidence for harnessing CH's regulatory role in milk fat synthesis and may hold implications for potential applications in the dairy industry and human nutrition.

## ACKNOWLEDGEMENTS

The authors would like to thank the Chinese Academy of Sciences, Kunming Institute of Zoology for the technical and platform support.

### Funding

This research was funded by the National Natural Science Foundation (Grant number: 31902152), the Scientific Research Foundation of Southwest Forestry University (Grant number: 112119), and the Ten Thousand Talent Plans for Young Top-notch Talents of Yunnan Province. The funders had no role in study design, data collection and analysis, decision to publish, or preparation of the manuscript.

### Grant Disclosures

The following grant information was disclosed by the authors:
National Natural Science Foundation: 31902152.
Scientific Research Foundation of Southwest Forestry University: 112119.
Ten Thousand Talent Plans for Young Top-notch Talents of Yunnan Province.

### Competing Interests

The authors declare that they have no competing interests.

### Author Contributions

- Mengxue Hu conceived and designed the experiments, performed the experiments, analyzed the data, prepared figures and/or tables, authored or reviewed drafts of the article, and approved the final draft.
- Lily Liu conceived and designed the experiments, performed the experiments, analyzed the data, prepared figures and/or tables, authored or reviewed drafts of the article, and approved the final draft.

### Data Availability

The raw measurements are available in the Supplemental Files.

### Supplemental Information

Supplemental information for this article can be found online at http://dx.doi.org/10.7717/peerj.16611#supplemental-information.

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
