# Peer review of "Choline regulation of triglycerides synthesis through ubiquintination pathway in MAC-T cells"

_PeerJ, doi:10.7717/peerj.16611_

## Round 0.1 · original submission · Major Revisions

Dear Drs. Hu and Liu:

Thanks for submitting your manuscript to PeerJ. I have now received two independent reviews of your work, and as you will see, the reviewers raised only minimal concerns about the study. Thus, these reviewers are optimistic about your work and the potential impact it will have on research studying lipid metabolism and regulation. Thus, I encourage you to revise your manuscript, accordingly, considering all of the concerns raised by both reviewers.

While the concerns of the reviewers are relatively minor, this is a major revision to ensure that the original reviewers have a chance to evaluate your responses to their concerns. There are many suggestions, which I am sure will greatly improve your manuscript once addressed.

Importantly, please ensure that all the information that is necessary to support your findings and observations is provide and methods are repeatable.

I look forward to seeing your revision, and thanks again for submitting your work to PeerJ.

Good luck with your revision,

-joe

·

Basic reporting

The article was written clear and unambiguous.

Experimental design

The authors investigated an important issue regarding to the regulatory mechanism of choline on milk fat synthesis in cows, hence the dairy milk industry should benefit from this study as well as the dairy farms. They emphasized that the choline's role in regulating triglyceride synthesis in MAC-T cells and its potential application as a feed additive for cattle, benefitting the dairy industry's milk production efficiency and economic outcomes. Which could be consider as an important finding, and it seems to me that it has relevance in the scientific world. In this sense, I recommend approving the manuscript. While, in my opinion there are a few cosmetics changes still needs for more enhancement the manuscript General comments.

Validity of the findings

1. The key words need to rearranged according to alphabetical order.
2. The abbreviations should be emphasized the whole name when first mentioned, please
3. (The bovine mammary gland epithelial cell line MAC-T originated from the frozen stock of our laboratory) Please more information needs to be provide about the MAC-T.
4. Some detergents and media like ( DMEM culture medium, fetal bovine serum (FBS), and peni cillin-streptomycin were purchased from GIBCO? The authors should put more information about this products, like the full scientific classifications , the company origin , country and state, the purity of the products etc. (This applies to all instruments, chemicals, or cultures that were used over the course of the experiment, Please
5. More information needs choline powder. Source and manufacture as I mentioned above
6. In my opinion some missing information in some techniques used in the experiment, the authors should reconsider this point
7. I have a question for the authors if the choline a promising product for changing the milk fat for better composition how would you apply it for the animals? Furthermore, what about the in vivo dose you will use? Also if animals took choline dose it not inter in the metabolic system and may be change his effect on the animal in vivo? Do you warranty this effect? Such points should be taken into considerations please
8. The conclusion is so long need to shorten, please, just to facilitate the reading process

·

Basic reporting

This manuscript aimed to investigate the regulatory mechanism of choline on triglycerides synthesis in MAC-T cells, with a specific focus on its potential association with high milk fat percentage in Zhongdian yaks' gut. The interest in this manuscript idea is significant, and it should be accepted if the authors make some revisions.

Experimental design

No comment.

Validity of the findings

No comment.

Additional comments

1. The abbreviations should be emphasized the whole name when first mentioned, please
2. Line 48, regent? Please revise it.
3. Provide more information about this reagents. For example, “DMEM culture medium, fetal bovine serum (FBS), and penicillin-streptomycin were purchased from GIBCO”. The authors should put more information about these.
4. Line 49 (The bovine mammary gland epithelial cell line MAC-T originated from the frozen stock of our laboratory) Please provide more information about the MAC-T.
5. Line 55, more information needs Choline powder? source and manufacture as I mentioned above
6. Line 63, the cells were seeded at a density of 2.5×105 cells per well, 105cells or 105 cells?
7. Line 70 to centrifugation, what about the conditions ? Please
8. Authors has carried out this experiment at the level of the cell itself, and one of the most important things that has been clarified is that this addition will positively affect the fat content of cow’s milk. I expected that such an experiment would be conducted at the level of the animal itself, and not just the cell level. To be an actual application for this product, I wish this will be consider in the authors future plan. In addition the cost and visibility study of using the choline should be taken also into consideration.
9. Figure 3, Improve the figure. Increase the size. Or add this figure as an annex in better resolution and size.
10. Line 225-238, this is not a choline specific pathway. Other molecules can activate this pathway. Describe and add a number to support this statement (Only as an example: in a multivariate analysis, choline showed 70% implicance in fatty acid synthesis)
11. Lines 301-320: I comprehend your conclusion. Yet, consider this: by introducing glucose to the other cells, it's plausible to observe an elevated glycolysis. Similarly, if I augment the concentration of medium or long chain fatty acids (such as oleic, stearic, linoleic, linolenic, etc.) in the mammary cell, one would anticipate an escalation in triglyceride synthesis, wouldn't you agree? And Why you choose choline?

---

## Round 0.2 · accepted · Accept

Dear Drs. Hu and Liu:

Thanks for revising your manuscript based on the concerns raised by the reviewers. I now believe that your manuscript is suitable for publication. Congratulations! I look forward to seeing this work in print, and I anticipate it being an important resource for groups studying studying lipid metabolism and regulation. Thanks again for choosing PeerJ to publish such important work.

Best,

-joe

·

Basic reporting

The article is written clearly and unambiguously. Meanwhile, references and sufficient field background/context were provided.

Experimental design

Methods were described with sufficient information to be reproducible by another investigator.

Validity of the findings

The conclusions were appropriately stated and connected to the original question investigated.

Additional comments

1. Table 1 should be moved to Supplementary Materials.
2. The type of note in table 1 was wrong.
3. Graphical Abstract can be provided.

·

Basic reporting

no comment

Experimental design

no comment

Validity of the findings

no comment

Additional comments

no comment